# Representations and Computations in Transformers that Support Generalization on Structured Tasks

**Yuxuan Li**                                                    *liyuxuan@stanford.edu*
*Department of Psychology*
*Stanford University*

**James L. McClelland**                                          *jlmcc@stanford.edu*
*Department of Psychology*
*Stanford University*

**Reviewed on OpenReview:** *https://openreview.net/forum?id=oFC2LAqS6Z*

## Abstract

Transformers have shown remarkable success in natural language processing and computer vision, serving as the foundation of large language and multimodal models. These networks can capture nuanced context sensitivity across high-dimensional language tokens or image pixels, but it remains unclear how highly structured behavior and systematic generalization can arise in these systems. Here, we explore the solution process a causal transformer discovers as it learns to solve a set of algorithmic tasks involving copying, sorting, and hierarchical compositions of these operations. We search for the minimal layer and head configuration sufficient to solve these tasks and unpack the roles of the attention heads, as well as how token representations are reweighted across layers to complement these roles. Our results provide new insights into how attention layers in transformers support structured computation within and across tasks: 1) Replacing fixed position labels with labels sampled from a larger set enables strong length generalization and faster learning. The learnable embeddings of these labels develop different representations, capturing sequence order if necessary, depending on task demand. 2) Two-layer transformers can learn reliable solutions to the multi-level problems we explore. The first layer tends to transform the input representation to allow the second layer to share computation across repeated components within a task or across related tasks. 3) We introduce an analysis pipeline that quantifies how the representation space in a given layer prioritizes different aspects of each item. We show that these representations prioritize information needed to guide attention relative to information that only requires downstream readout.

## 1 Introduction

Since their introduction, transformer-based models (Vaswani et al., 2017) have become the new norm of natural language modeling (e.g. Brown et al., 2020; Devlin et al., 2018) and are being leveraged for machine vision tasks as well as in reinforcement learning contexts (e.g. Chen et al., 2021; Dosovitskiy et al., 2020; Janner et al., 2021). Transformers trained on large amounts of data under simple self-supervised, sequence modeling objectives are capable of subsequent generalization to a wide variety of tasks, making them an appealing option for building multi-modal, multi-task, generalist agents (e.g. Bommasani et al., 2021; Reed et al., 2022). In particular, large-size transformer-based language models have shown impressive generalization beyond language tasks, in arithmetic tasks, code generalization, and sometimes reasoning and planning problems (e.g. Bubeck et al., 2023; Chowdhery et al., 2022; OpenAI, 2023). These findings have led to ongoing work that evaluates the broader reasoning capabilities in these models (e.g. Binz and Schulz, 2022; Dasgupta et al., 2022; Dziri et al., 2023).

Central to this success is the ability to represent each part of the input in the context of other parts through the self-attention mechanism. This may be especially important for task objectives with naturalistic data such as next-word prediction and image classification, which requires the exploitation of nuanced context sensitivity across high-dimensional inputs. Interestingly, analyses of transformer-based language models suggest that they can acquire knowledge of syntactic structures without being explicitly trained to do so (Linzen and Baroni, 2021; Manning et al., 2020).

Despite success in learning large-scale, naturalistic data, how well transformer models support systematic generalization and their internal computation remains to be better understood. Recent work has demonstrated that transformer-based language models struggle with longer problems and fail to robustly reason beyond the training data (Anil et al., 2022; Razeghi et al., 2022). Different architectural variations have been proposed to improve length generalization in transformers, highlighting the role of variants of position-based encodings (Csordás et al., 2021a;b; Ontanón et al., 2021; Press et al., 2021). Indeed, whether neural networks will ever be capable of systematic generalization without building in explicit symbolic components remains an open question (Fodor and Pylyshyn, 1988; Smolensky et al., 2022).

Our goal in this paper is to explore what solutions transformers develop to solve structured tasks, and assess the roles attention heads and token representations play in supporting such solutions. We approach this question by training a causal transformer to perform a set of algorithmic operations, including copy, reverse, and hierarchical grouping and sorting tasks. We explicitly seek the minimal transformer that would reliably solve these tasks either in a single-task setting or in a task-cued, multi-task setting. We then analyze such minimal solutions through attention ablation and representation analysis. Exploring how a transformer with no pre-defined task-aligned structure can adapt to structures in these algorithmic tasks provides a starting point for understanding how self-attention can tune to structures in more complex problems, e.g., those with the kinds of exceptions and partial regularities of natural datasets, where the exploitation of task structures may occur in a more approximate and graded manner. Our main contributions are:

1. We introduce a simple label-based encoding method that samples random integer labels from a label pool larger than the maximum sequence length and uses encodings of these labels in place of position-based encodings. We show that this method helps transformers achieve faster in-distribution learning and strong length generalization compared to position-based encoding methods, and that learnable label encodings come to represent sequence order in tasks requiring sequence order information.

2. We show that, in support of robust and generalizable solutions to the algorithmic tasks, two-layer causal transformers develop signs of systematic decomposition within tasks and exploitation of shared structures across tasks. For example, when learning to solve a task that sorts items by item features, two-layer transformers learned a two-step algorithm where attention patterns focused on one item group (e.g, all circle items) at a time. When two-layer models learn multiple tasks at once, each attention head contributes to multiple related tasks (e.g., all tasks that rely on item features).

3. We propose a novel representation analysis pipeline, using singular value decomposition and variance partitioning to quantify how the representations evolve across layers. Using this method, we find that transformers use sequence contextualization to re-weight task-relevant information to dominate the representation space. For example, when only item feature information is needed to sort the sequence, item representations in the intermediate layer prioritize representing item features over item order, while in tasks where only order is required for sorting, item order is prioritized over item features.

## 2 Method

**Dataset**. We created an item pool covering all combinations of 5 shapes, 5 colors, and 5 textures and generated a sequence dataset by sampling 100k sequences of 5–50 items randomly selected from the item pool. The tasks we used to train the models are shown in Figure 1A. Each task corresponds to one of the following rules, which relies on item feature and/or item order information to rearrange items in the input sequence (grouping or sorting by item feature is with respect to a pre-defined feature sort order, e.g., circles < squares < pentagons, or red < purple < blue):

COPY (C): copy the input sequence.
REVERSE (R): reverse the input sequence.
GROUP[SHAPE] (G[S]): group the items by shape, preserve the input order within each shape group.
GROUP[COLOR] (G[C]): group the items by color, preserve the input order within each color group.
SORT[SHAPE,COLOR,TEXTURE] (S[S,C,T]): sort the items first by shape, then by color, then by texture.
SORT[COLOR,SHAPE,TEXTURE] (S[C,S,T]): sort the items first by color, then by shape, then by texture.

We instantiated the token vocabularies as onehot and multihot vectors. Task tokens were onehot vectors with the corresponding task category set to one, with one additional task dimension corresponding to the end-of-sequence (EOS) token. Item tokens were multihot vectors whose units indicated its value in each feature dimension (equivalent to concatenated onehot feature vectors). As such, the model receives disentangled feature information in the input, though in principle it can learn to disentangle feature information given onehot encodings for each unique item.

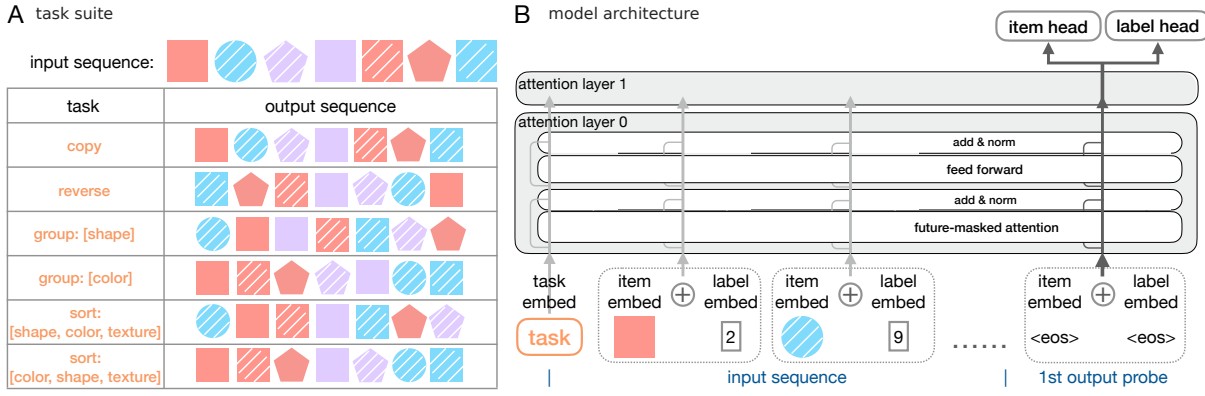

Figure 1: Task and model design.

**Label-based encoding**. Models using position-based encodings trained with sequences up to length $L$ encounter an out-of-distribution problem when tested on longer sequences (in either absolute or relative position-based settings). We conjectured that this problem arises because useful internal representations have not been learned for positions beyond those encountered in sequences up to length $L$. To overcome this problem, we introduce label-based encoding, which instead pairs items in each sequence with random integer labels sampled from a set of labels larger than sequence positions. Specifically, we initialize a pool of $K$ available labels. For a sequence of $N$ tokens, we randomly sample $N$ out of $K$ labels (with $K > N$) and use their corresponding label encodings to encode the sequence. In our experiments, the sampled labels are always assigned in ascending order to the items in each sequence. Though this scheme does not capture exact position information, it is sufficient to convey sequence order information, since each successive item is always associated with a larger integer label than its predecessor. The sampled labels could, alternatively, be assigned to items in random order, in which case they would function simply as arbitrary item labels. Either way, randomly sampling labels allows the model to learn to form representations during training for a set of labels larger than the maximum training sequence length. As we show below, this scheme supports length generalization in the tasks we consider with very mild degradation as a function of sequence length up to the number of available integer labels.

In our experiments, we use learnable embedding weights as the label encodings, and contrast the label encoding method with sinusoidal and learnable embeddings based on fixed position indices. Concurrent work also explored the same method as our label-based encoding method (termed randomized positional encodings, see Ruoss et al. 2023) and tested it with other types of embeddings. In all reported results, we used the 100k sequences mentioned above and pre-generated item labels from a range up to the maximum generalization length in the dataset ($K = 50$). In practice, the labels for each sequence can be sampled online and from a larger range to accommodate generalization to even longer sequences.

**Model**. The main model architecture is shown in Figure 1B. Each input sequence consisted of a task token, a sequence of the paired item and label tokens, and an EOS token. The input tokens were first embedded in the model's latent representational space through a set of embedding layers depending on the token type (task, item, or label). The item and label embeddings were then added to form a composite item embedding. These embedded tokens were fed into a causal transformer, which contained layers of alternating future-masked attention sublayers and MLP sublayers. Residual connections and layer normalization were applied after each sublayer as in Vaswani et al. (2017). We tested architectural variations in the number of attention heads in different layers of the model while controlling for the total number of learnable parameters (see detailed hyperparameters in Appendix B). The EOS token served as the first probe for tokens in the output sequence. The state of the probe item at the output of the causal transformer was passed through two linear heads to predict the next output token (the task token, or an item and its associated label).

**Training and evaluation**. All models were trained using teacher forcing, i.e., we always feed the model the correct prior sequence of tokens as inputs during training. The training set consisted of all sequences of lengths 5 to 25 in the dataset ($\sim$46k), and models were evaluated for length generalization on sequences of lengths 26 to 50 ($\sim$54k) and novel sequences in the training length range. We trained models in both single-task and multi-task settings. In both cases, the output sequence consisted of the correctly ordered items and their labels given the task being trained, followed by an EOS token. In single-task learning, we did not include the task token in training or evaluation. In multi-task learning, the task token was used and the models were trained to first output the task token before predicting the output sequence. The training sequences used in multi-task learning were the same length 5–25 sequences from the dataset, but the corresponding output sequence changed depending on the task.

The models were trained using softmax cross-entropy losses on the prediction of feature classes, labels, and task/EOS categories for tokens in the output sequence. Item predictions were treated as average feature prediction accuracy, i.e., if the model predicted 2/3 features correct, its token-level item accuracy is 2/3. Training stopped at 32k gradient updates for single-task models and 38k gradient updates for multi-task models. Below, we report both token-level and sequence-level accuracy, under both teacher forcing and top1 rollout (i.e., greedy decoding). We ran four random seeds for each task × architecture pair. Unless otherwise specified, results were taken from the checkpoint with the highest generalization accuracy within each seed. Error shades and error bars indicate the standard error of the mean across models.

## 3 Results

### 3.1 Label-based encoding supports length generalization and order representation

We first demonstrate the effectiveness of the label-based encoding method. We trained two-layer, single-headed models on the SORT[SHAPE,COLOR,TEXTURE] task in the single-task setting. Using the label-based encoding method, these models were able to achieve near-ceiling accuracy on training sequences and generalize to longer sequences with comparable performance, but models trained with sinusoidal or learnable position encodings performed worse across both training and generalization sequences (Figure 2A; also see quantitative results in Table S2).

We found that label embeddings learn different representations depending on whether sequence order information is needed to solve the task. In the SORT[SHAPE,COLOR,TEXTURE] task, item positions in a sequence do not affect their output position. However, in the COPY, REVERSE, and GROUP tasks, the model needs to rely on item order in the input sequence to correctly sort the sequence. We thus compared the label embeddings learned by the single-task model trained on the SORT[SHAPE,COLOR,TEXTURE] task and the

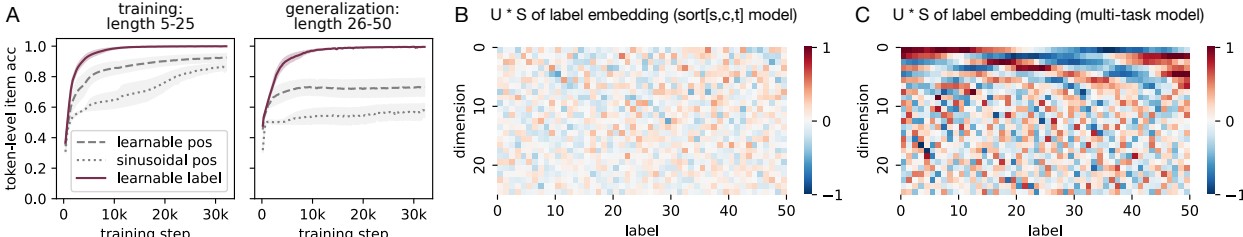

Figure 2: **A**. Label-based encoding enables fast learning and strong generalization compared to other methods. **B** and **C**. Left singular vectors (scaled by singular values) from the singular value decomposition of the learned label embeddings. The decomposed top dimensions of the label embeddings in the multi-task model show Fourier modes from low to high frequency.

multi-task model that learned to solve all six tasks. We used singular value decomposition to inspect the most prominent modes in each set of embeddings. The label embeddings in the multi-task model learned a strong order representation, marked by Fourier modes of increasing frequency in the top decomposed dimensions (Figure 2C). This is typical for a chain- or ring-like structural form (Saxe et al., 2019). In contrast, the label embeddings associated with the single SORT task model resemble more independent labels and do not contain explicit order information (Figure 2B). In this case, length generalization still occurred with these unstructured representations as input order information is not needed to specify the sort order, and the model relied on developing structured attention patterns to solve the task (discussed below).

## 3.2 Successful solution relies on multiple layers and shared attention heads

Label embedding is not the only factor that contributes to finding a robust solution for these tasks that generalize to longer sequences. As shown in Figure 3, architectural variations in the number of model layers and the placement of multiple attention heads can also have significant impacts on the resulting performance. When learning only the SORT[SHAPE,COLOR,TEXTURE] task, two-layer, single-headed models (designated as [1,1]) were much better than single-layer models with either one or two attention heads (designated as [1] and [2], Figure 3A). The single-layer models were able to exploit some correlations between items and output positions (e.g., that orange-empty-circle always came first, and blue-striped-cross always came last), but they failed to sort items in the middle positions (Figure 3A1, right). In contrast, a single-layer, single-headed model was sufficient to learn the COPY or the REVERSE task (see Table S2 in Appendix C), suggesting that multiple layers strongly benefit successful learning of multi-level problems.

When only appended with a task token, the two-layer, single-headed model that succeeded in single-task learning was unable to learn all six tasks at once. Instead, two-layer, multi-headed models achieved good performance and were able to generalize all tasks to longer sequences (Figures 3B; see quantitative performance in Table S3). In addition, models with multi-headed attention in the second layer generally achieved higher accuracy across all tasks compared to models with multi-headed attention in the first layer (Figure 3B). Models with multi-headed attention in the second layer were even comparable with models that have multi-headed attention in both layers. This signals that a bottlenecked architecture where the attention head is shared in the first layer may be particularly suited for multi-task learning in these tasks, considering that some tasks share the first-level grouping feature.

In both single-task and multi-task settings, the top-performing two-layer models showed some degradation in sequence-level accuracy as a function of sequence length, but the failures on longer sequences were not catastrophic, as these models scored very well on longer sequences when up to 5% prediction errors were allowed (Figures 3A2 and 3C2).

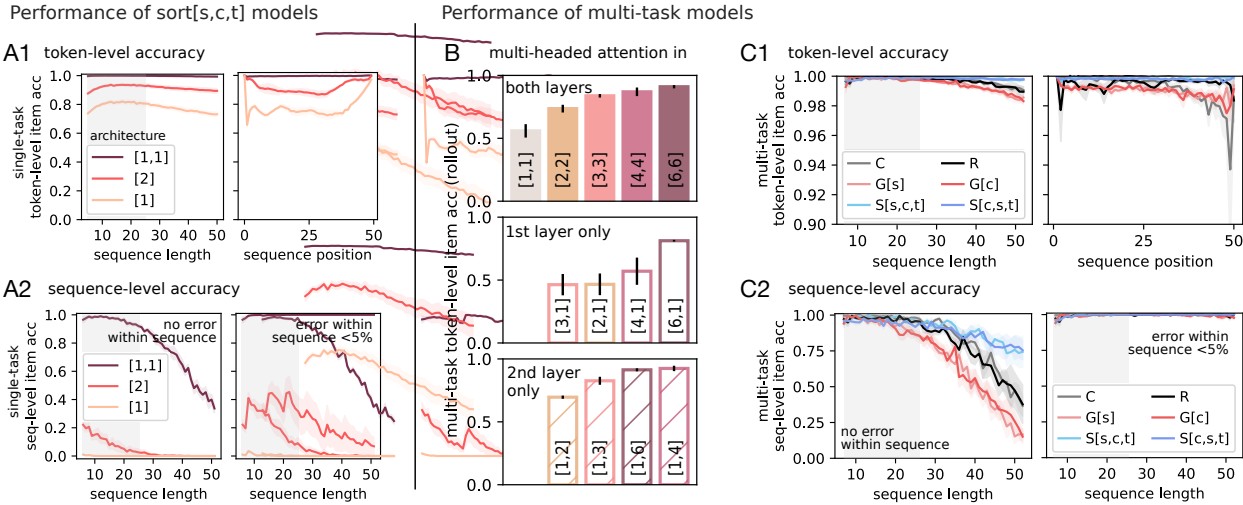

Figure 3: Performance of single SORT[S,C,T] task models and six-task models. **A1** and **C1**. Token-level item accuracy over sequence lengths and sequence positions. **A2** and **C2**. Sequence-level item accuracy – the proportion of sequences where models make no error (left) or make less than 5% errors (right) within sequences. **B**. Multi-task models' token-level item accuracy from top1 rollout on 1.2k length generalization sequences across all tasks (200 per task). Models are arranged in ascending order of performance. See more accuracy under rollout in Figures S1. Legends in A and B indicate the number of attention heads in each model layer. Legends in C indicate the six tasks. Gray shades indicate the range of lengths used in training. Results in A and C within the training length range are from novel sequences.

### 3.3 Early layers transform and reduce the task to support repeated computation in later layers

Are these models performing different computations across layers to solve multi-level problems? We closely inspected the attention patterns of the top-performing models to understand the solutions they found. As discussed below, this analysis revealed distinct multi-stage processing across successive attention layers. In both single-task and multi-task settings, we found that earlier layers tended to transform token representations such that subsequent layers can deploy similar computation across items and across tasks.

**Within-task decomposition in the single-task model**. Figures 4A and 4B show the learned solution process of the top-performing two-layer, single-headed model that successfully solved the SORT[SHAPE,COLOR,TEXTURE] task in the single-task setting. When given a probe item after the presentation of the input sequence, the attention head in the first layer distributed attention to the unsorted items in the same shape group as the current probe item. The attention head in the second layer then almost exclusively attended to the target next item in preparation for feature and label readout. This pattern appeared robustly across sequences and across different seeds. Interestingly, there was also an increase in the attention weights to the EOS token as the model received probe items towards the end of each shape group (Figure 4C). This attention to EOS increased to similar degrees in early or late shape groups, again suggesting that the model learned to systematically process items within each shape group, even though generating the EOS token was only relevant after sorting all items.

We found a similar hierarchical attention pattern in two-layer, single-headed models trained to perform the GROUP[SHAPE] task (see Figure S2C in Appendix A). In contrast, the single-layer models learning the SORT[SHAPE,COLOR,TEXTURE] task displayed some attention to subsequent items in the output sequence but lacked consistent structures (see Figures S2A and S2B). This could be due to the burden for the attention head(s) within a single layer to implement a mixture of item contextualization and readout of the correct target information, and reflects an advantage of the two-layer models in providing the capacity to develop a structured, multi-stage solution.

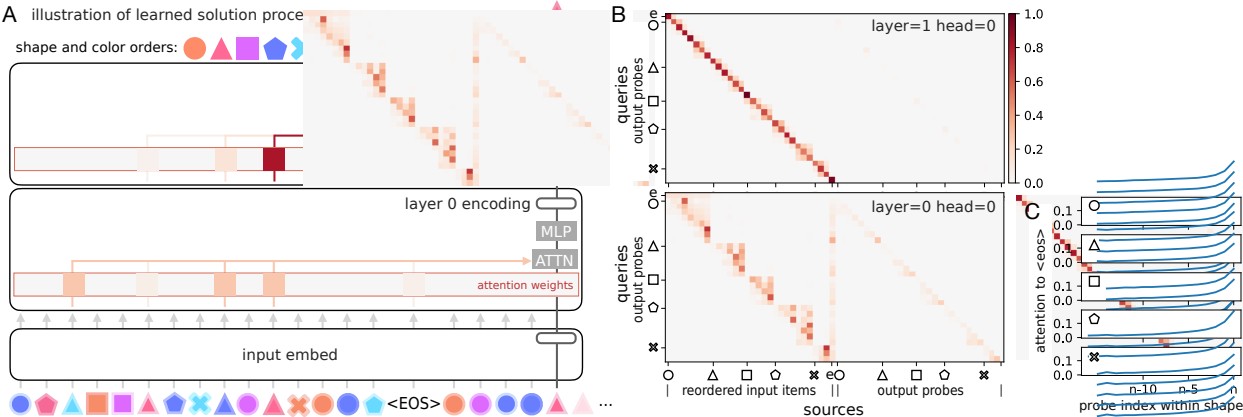

Figure 4: Single-task model's solution to SORT[SHAPE,COLOR,TEXTURE]. **A**. An illustration of a single step in the learned solution process across layers on an example sequence, equivalent to one row in B. Item textures are visualized as edge colors. **B**. Attention maps for predicting all target items in a generalization sequence. Items in the input sequence were reordered to match their output order for visualization purposes. The five shapes mark the beginning of each shape group. Label e indicates the EOS token. **C**. Attention to EOS increases as models generate items in each shape group. Results in C were aggregated across 1k generalization sequences and across all four two-layer models.

**Shared computation across tasks in the multi-task model**. Consistent with the single-task models, multi-task models also developed a distinct multi-stage solution. Strong performance from models with fewer attention heads than there are tasks already hinted that the model could be exploiting shared processing across related tasks. Indeed, the attention heads in the top-performing model (with one attention head in the first layer and four attention heads in the second layer) show little task selectivity: ablating each attention head resulted in impaired performance in multiple tasks, and the drop in performance is usually to similar degrees for each pair of related tasks (Figure 5A; see ablation results from other models in Figure S3 in Appendix A). The attention heads also contributed equally to target feature and label prediction (Figure 5A, left and middle). Some heads appeared less important for accurately predicting the next item under teacher forcing, but the performance score here is misleading. In fact one of the heads (L1-H1) was crucial for shifting from the current top-level group to the next group; its ablation led to a systematic failure to generate the correct first item in the next group. Teacher-forcing the first item of each group provided this information to the model, allowing it to sort the remaining items in each group correctly, but without this head the accuracy for the multi-level problems is significantly impaired under top1 rollout (Figure 5A, right). We next discuss the weight patterns observed in these attention heads.

The attention head in the first layer was heavily relying on the task token to transform item representations from the same input sequence differently under each task (Figure 5C, L0-H0). As a basis of this task-dependent contextualization, the pairwise similarities between the learned task embeddings indeed reflected similarities on the piece of item information (feature or label) that is used for sorting the items in each task (Figure 5B).

Although the attention pattern in the first layer of the multi-task model differs from that in the two-layer single-task models, it effectively allowed the attention heads in the second layer to perform similar processing across different tasks (see Figure S4 for full attention maps across all heads). For example, some heads in the second layer collectively attended to the target next item in the input sequence (marked by the diagonal attention weights distributed across multiple heads), together resembling the attention pattern in the second layer of the single-task model (Figure 5C, L1-H0 and L1-H3). The multi-task model also developed a different signature of within-task decomposition. As mentioned above, one attention head in the second layer played a crucial role in correctly shifting to the next top-level feature group. Its attention pattern showed consistent attention to the first item in the next feature group as the model generated target items in the GROUP and SORT tasks (Figure 5C, L1-H1). These results suggest that the multi-task models are simultaneously transforming tasks into similar ones and reducing tasks that are multi-level.

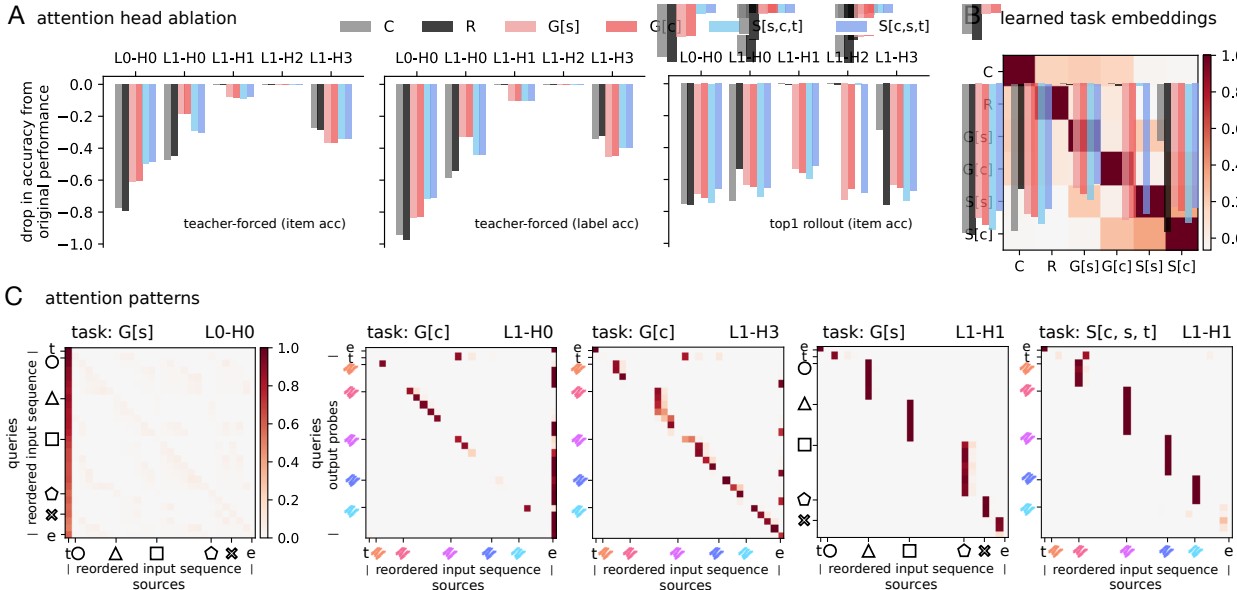

Figure 5: Unpacking the solution process of the multi-task model. **A**. Token-level accuracy across all tasks from ablating single attention heads. **B**. Pairwise similarities in the learned task embeddings. **C**. Partial attention maps from a generalization sequence in selected attention heads (see Figure S4 for full attention maps in all heads). Items corresponding to the input sequence were reordered to match their output order for visualization purposes. The five shapes and colors mark the beginning of each feature group. Label t indicates the task token. Label e indicates the EOS token. In A and C, results were taken from the top-performing model (architecture: [1,4]). Results in B were aggregated across the five top-performing models (across architectural variations).

### 3.4 Token contextualization prioritizes task-relevant information in the representation space

How are token representations transformed as a result of and in support of the structured attention patterns in these models? As shown in Figure 6A, there were systematic discrepancies in how accurately and quickly the multi-task model predicted output features and labels under different tasks, suggesting that the model prioritized representing the sort-relevant information more robustly over information only for downstream readout. We next examined the transformation of item representations over layers to confirm this hypothesis and to explore the extent to which the tasks are functionally treated as one canonical item rearrangement task. As we discuss in more detail in the next few paragraphs, the results reveal that a stronger representation of sort-relevant information in the model reflects a fundamental feature of its learned solution: when contextualizing items differently under different tasks, the model re-weighted task-relevant information to dominate the representation space in support of downstream shared computation.

**Partitioning variance in the representation space**. We devised a representation analysis pipeline to quantify how different pieces of information compete in the representation space. For a given layer, we performed singular value decomposition (SVD) on the collection of representations associated with the output state of each item at that layer. This included all 6250 possible token embeddings (125 items × 50 labels) in the input embedding layer and contextualized item representations from sampled sequences in subsequent attention layers. Representations in each layer were first zero-centered separately for each unit in the layer, so that the dimensions of the SVD correspond to dimensions from principal component analysis (PCA). Then for each dimension in the SVD, we computed the proportion of the variance in its unit-variance left singular vector that was explained by each feature type (color, shape, or texture) and by label information. These explained portions of variance were then scaled by the variance associated with that dimension. This allowed us to quantify which aspect of item information dominated each dimension, and then to sum these quantities across dimensions to assign a total variance-explained ratio to each piece of item information in a layer's representation space.

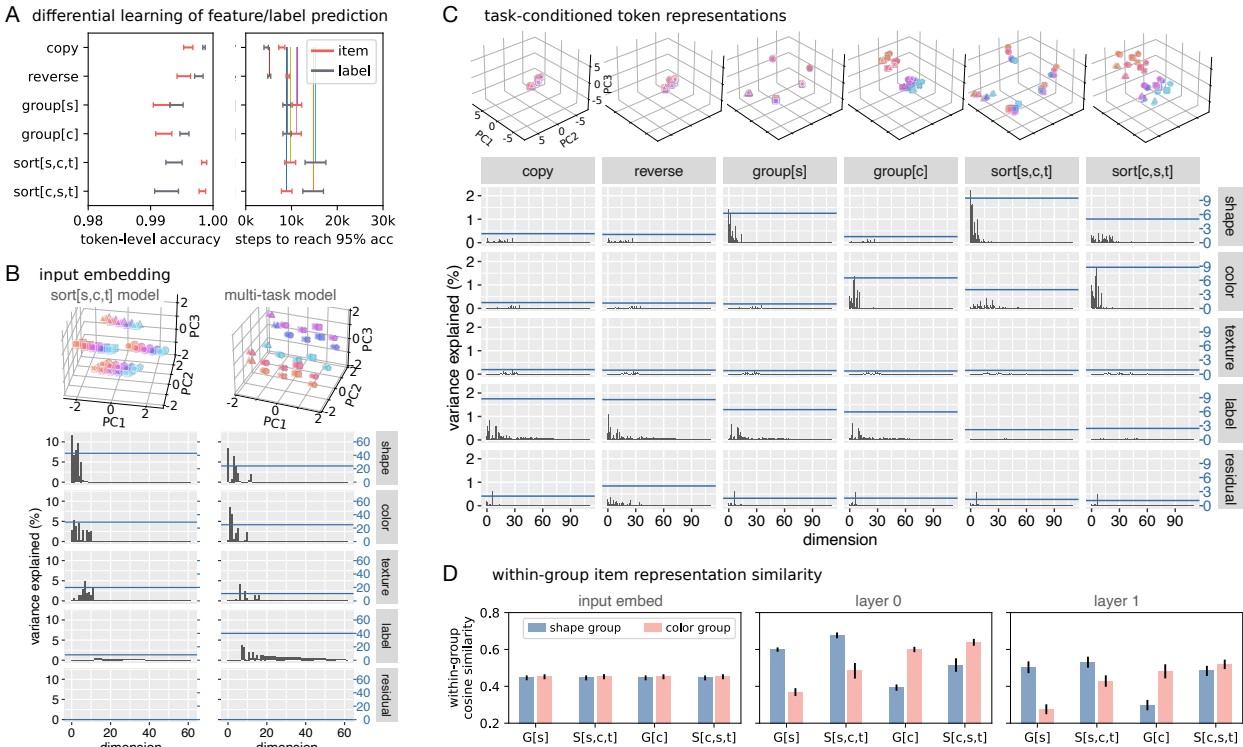

Figure 6: Models prioritize representations for sort-relevant information. **A**. Token-level feature and label prediction accuracy on generalization sequences (left) and the time taken to reach 95% accuracy (right). **B**. Learned input embeddings from the top-performing single-task model (trained on S[s,c,t]) and the multi-task model. Top, the lower-dimensional projection of input item feature embeddings. Item textures are visualized as edge colors. Bottom, variance explained in each dimension of the full decomposition of all 6250 (125 items × 50 labels) input embedding representations. **C**. Representations of input and probe items in layer 0 of the multi-task model. Top, the lower-dimensional projection of these representations. Bottom, the variance-explained profile. **D**. Mean within-group pairwise similarity of input item representations in the multi-task model. In B and C, blue lines represent the total variance explained in a category across all dimensions. Weaker dimensions beyond those displayed are excluded for visualization purposes.

**Feature/label information in the input embeddings**. Our results suggest that item representations as early as in the task-agnostic input embeddings are sensitive to the general relevance of different pieces of information. Figure 6B shows the variance-explained profile across the decomposed dimensions of the input embeddings. The multi-task model weighted shape, color, and label information (more relevant for sorting the items) more than texture information (mostly for readout), with shape and color information occupying the top-most dimensions. This variance-explained profile stands in stark contrast to that of the input embeddings in the single SORT task model, where item feature information together dominated the top 12 dimensions of larger variance, and label information is strictly separated from feature information and occupied weaker dimensions.

**Task-conditioned token representations in the intermediate layer**. Applying the same method to items sampled from 200 length-30 sequences, we obtain a full view of how representations evolved under each task in layer 0. In this layer, the output state of the source items and the probe items jointly reflect a re-weighting of the sort-relevant information, with the total amount of variance explained by each feature or label reflecting their relative importance for sorting items in any given task (Figure 6C). For example, in the GROUP[SHAPE] and SORT[SHAPE,COLOR,TEXTURE] tasks, shape information occupies the top-most dimensions, whereas color information occupies the top-most dimensions in the GROUP[COLOR] and SORT[COLOR,SHAPE,TEXTURE] tasks. Meanwhile, label information was brought to top dimensions

in the four tasks that rely on item order to sort the items (COPY, REVERSE, and the GROUP tasks). The stronger representations of the grouping features in the GROUP and SORT tasks resulted in an increase in the within-group item similarity in the intermediate layer, i.e., a stronger sense of item clustering following contextualization (Figure 6D). Notably, within-group item similarity also increased for the secondary grouping feature in the SORT tasks to reflect an elevated but relatively weaker sense of item clustering.

We used a cross-decoding analysis to explore the possibility that the model is mapping different feature information to a shared functional encoding for related tasks. We trained a linear classifier to extract shape category from layer 0 probe item representations under the GROUP[SHAPE] task, and tested whether this classifier perfectly transferred to classifying item color in the GROUP[COLOR] task (chance level: 20%). The results show that such a classifier was above chance but only somewhat effective when classifying the color of the probe items in the GROUP[COLOR] task (accuracy: 57%), equaling the transfer to decoding color category in the SORT[SHAPE, COLOR, TEXTURE] task (accuracy: 59%). Instead, the classifier best transferred to extracting shape category in the SORT[SHAPE, COLOR, TEXTURE] task (accuracy: 100%). These results indicate that the model still largely preserved feature-unique information and may only be partially transforming these functionally-equivalent features to a shared encoding. Although specific attention heads could in principle rely more on the shared encoding information, at the level of item output states in this layer, it seems that the model remains sensitive to the different tasks and is only partially transforming related tasks into the exact same one.

**Probe and target representations in the final layer**. The re-weighted representational characteristics of the probe items in layer 0 carried over to layer 1, although representation for the target item dominated the layer 1 representation space (Figure S5). For this analysis, we turned to a set of 2000 randomly-sampled two-item sequences to decouple feature correlation between the probe and target items. Here, the variance-explained profile of the layer 1 target item representations is more similar than that in layer 0, suggesting that the model projected target item representation into a more shared space for readout. Notably, the model performance on these two-item sequences was lower than on longer sequences. The performance on the single-level COPY and REVERSE tasks remained at 99% and 98%, but performance on the multi-level tasks (GROUP and SORT tasks) dropped to around 90%. This stark contrast in performance reflects that two-item sequences are challenging for the model, as the model developed a strategy to rely on the multi-level grouping structure to identify the sort order of a given sequence, which is more suitable for sorting longer sequences.

## 4    Related Work

There is a growing interest in analyzing small models in more controlled task settings to better understand the capabilities and detailed computations in transformers. For example, Power et al. (2022) explored learning and generalization dynamics in two-layer causal transformers learning binary operations, and Elhage et al. (2021) explored mechanistic interpretability in one- and two-layer transformers without MLP sublayers. Our work contributes to these efforts in beginning to shape some understanding of the representations and computations transformers can develop to support generalization on structured tasks.

Recent work examining systematic generalization in transformers or pre-trained language models highlighted that length generalization remains a challenge and observed that positional encoding can have a significant impact on the extent to which models can systematically generalize (Anil et al., 2022; Csordás et al., 2021a; Delétang et al., 2022; Ontanón et al., 2021). Many types of architectural modifications have also been proposed to help transformers achieve better length generalization, including different ways to represent positional information (e.g. Csordás et al., 2021b; Dehghani et al., 2018; Press et al., 2021; Su et al., 2021). Our label-based encoding method adds to this effort by demonstrating the potential of formulating sequence modeling tasks under a more general item-label binding approach rather than item-position binding. Concurrent to our work, Ruoss et al. (2023) developed randomized position encoding (equivalent to our label-based encoding method) and showed its advantage over a variety of position-based encodings. We view our work as complementary to theirs, in that they demonstrated strong length generalization to longer sequences (with up to 500 elements) and over a range of tasks, while we have uncovered how these labels are differentially encoded and used in service of performing different tasks.

Outside of the context of transformers and language models, the use of synthetic, algorithmic tasks has enabled much understanding of the core capabilities of many models (e.g. Graves et al., 2014). There has also been some interest in performing algorithmic reasoning with neural networks for its own sake (Veličković and Blundell, 2021). Correspondingly, Veličković et al. (2022) recently proposed a benchmark for algorithmic reasoning, and evaluated a variety of graph neural network architectures, all of which struggled to extrapolate algorithms to longer or larger inputs. Our work shows that self-attention has the potential to adapt to structures in the sequence and find reliable solutions to algorithmic tasks.

The representation analysis method we employ here builds on the use of singular value decomposition in Saxe et al. (2019) to visualize multi-dimensional representational structure and to relate the learning dynamics in deep neural networks to the strengths of the singular dimensions in the SVD of the training data. We have extended this analysis by partitioning the variance in each singular dimension as associated with representing item feature and order information, then summing the variance across dimensions to shed light on how different kinds of information are prioritized in solving tasks with different structures. Below we consider how the approach in Saxe et al. (2019) and related more recent work might be further extended to address representation and learning dynamics in more naturalistic tasks and datasets.

Ablation experiments and a variety of analysis methods have been applied to understand the role of the attention mechanism in NLP tasks as well as in transformer-based vision models (Chefer et al., 2021; Manning et al., 2020; Michel et al., 2019; Voita et al., 2019). However, it is often debated to what extent attention weights afford model interpretability in these settings (Jain and Wallace, 2019; Wiegreffe and Pinter, 2019; Vashishth et al., 2019), especially considering head redundancy and the difficulty in correctly attributing relevance over high-dimensional inputs. We show that at least in simple settings, attention heads can exhibit some level of interpretability consistent with known task structures. Similar methods have also been applied to understand unit-level and layer-level computations that support multi-task computation in small-scale RNNs (Driscoll et al., 2022; Yang et al., 2019), which revealed some patterns that are consistent with our findings here, as we discuss below.

## 5 Discussion

We sought to understand how transformers can solve a set of highly-structured algorithmic tasks and systematically generalize. We presented two-layer causal transformers that can learn copying, reversing, and hierarchical sorting operations that generalize to sequences longer than seen during training. Detailed analysis of the attention patterns suggest that these models tended to rely on similar computation to sort items in different feature groups (within-task decomposition) and also to sort items in different tasks (across-task shared computation). Using a novel representation analysis pipeline to quantitatively partition variance in the representation space, we also find that the attention layers learned to re-prioritize task-relevant information in a task-dependent manner, which helps attention heads in subsequent layers multitask.

We highlight that the use of a random sample of ordered position labels was key to enabling our models to learn robust solutions to these tasks and generalize the learned tasks to longer sequences. The key insight is to sample random labels to communicate sequence order information instead of relying on fixed position indices during learning. This simple extension exposes models to a large range of possible labels during training so that longer sequences can be encoded with familiar labels, and is shown to be effective both in our tasks and in a range of different algorithmic tasks (Ruoss et al., 2023). Compared to this approach, learnable encodings tied to fixed sequence positions resulted not only in poor length generalization, but also in slower learning on the training sequences in our tasks. We suggest that this likely occurred because the models experienced relatively fewer opportunities to learn the position encodings that only appeared in longer sequences (e.g., position encodings 1-5 always appeared in all training sequences, but position encodings 20-25 appeared much less frequently). This poses a problem even in cases where position information is not required and the position labels only serve as arbitrary labels (as in the SORT tasks in our task suite). Sinusoidal positional encodings are known to be limited in their ability to support length generalization (Ontanón et al., 2021; Csordás et al., 2021a). In our setting, the demand to read out the corresponding item positions, which is not common across NLP tasks, may have further limited the usefulness of this scheme.

Our findings and those of Ruoss et al. (2023) are important in indicating that length *per se* is not an intrinsic limiting factor in transformers – up to, of course, the total length of the transformer's context memory. This makes sense given that the query-based attention mechanism treats all positions equivalently except insofar as it is sensitive to the actual positional information encoded in the keys, queries and values. It is worth noting that using a random ordered subset of position labels would not represent absolute item distance information. Such a scheme can, however, capture sequence order, which appears to be sufficient for our tasks and those of Ruoss et al. (2023). This scheme may also prove sufficient in natural tasks, and this will be an important question to explore in future research. For example, in natural language, it may be sufficient to represent the relative proximity of items in the context rather than their absolute distance, since constituents such as noun phrases can be of arbitrary length. Of course, it may be possible to combine label-based encoding and other distance-preserving position encoding schemes to allow a model to achieve length generalization while remaining sensitive to absolute token distances. We further note that transformers can acquire sensitivity to position information without any explicit positional information (Haviv et al., 2022). This and other recent work suggest that additional research is needed to fully understand the capacities and relative merits of the full range of approaches to providing sequence and positional information to transformers and other architectures.

Our synthetic task suite and minimal-architecture approach have allowed us to elucidate the computations and representations used to solve our tasks in the models we have studied. Like others, we are ultimately interested in understanding models trained on naturalistic tasks, which have rich and only partially systematic structures and are more challenging to fully understand. In our view, the study of synthetic tasks and naturalistic tasks can mutually inform each other, and we hope our work leads to more discourse between research in these complementary domains. Specifically, we hope that the results we show here provide useful insights and analysis tools that can be applied and extended to enhance our understanding of the computations and mechanisms that emerge in larger models. Some of our findings already show convergence with other studies using more complex tasks or different model architectures. For example, our findings on the contributions of individual attention heads in solving our algorithmic tasks echo results from analyses of language models. Single attention heads are rarely responsible for a particular task or syntactic relationship and can often appear redundant (Manning et al., 2020; Michel et al., 2019; Voita et al., 2019). We do see some degree of selectivity, with conceptually distinct task components shared across a subset of attention heads. Recent work studying multi-task computation in RNNs has similarly found that multi-task learning led to the exploitation of reusable computation across related tasks (Driscoll et al., 2022; Yang et al., 2019). Interestingly, these recurrent models also exhibited high degrees of task-selectivity at the level of individual units in the hidden layer. The degree of task-selectivity in different architectural components may vary depending on the inductive biases of different model families and the tasks being learned. Future work is needed to fully characterize the degree of possible computational modularity in multi-headed attention, and to confirm whether our specific findings generalize to broader settings.

The representational analysis approach we have applied here is a stepping stone toward uncovering how gradient-based learning leads to the development of structured solutions in neural network models. The analysis tools we have extended from Saxe et al. (2019) were developed in the context of exact analytic solutions to the non-linear learning dynamics in linear multi-layer networks in terms of the strengths of the singular modes that characterize the structure of the training data. Saxe et al. (2022) have recently extended this analytic reduction to be used to analyze a wide range of feed-forward network architectures. In future work, we foresee the possibility of using this approach to clarify how the gradient signals flowing through our transformer network serve to prioritize task-relevant information, and even how they allow the task embeddings to modulate this prioritization process in a task-dependent manner. We also expect the approach to be useful for uncovering the priority given to different kinds of information in more naturalistic task settings, including image-based tasks or natural language and reasoning tasks in transformer-based language models. Saxe et al. (2022) have taken a small step in this direction. One natural application of the approach will be to understand the relative prioritization and synergistic integration of in-context vs. in-weights information as explored experimentally in Chan et al. (2022). We consider the prioritization and integration of these two sources of information to be central to understanding the general ability of transformers to behave as multi-task systems.

The flexibility to learn and perform multiple tasks is a key desired capability for machine learning and artificial intelligence. Our work here provides insights into the structured within-task and cross-task computations that stacks of attention layers develop when learning highly structured tasks. Recent work has explored multi-task learning in transformers at scale and achieved impressive results (Lee et al., 2022; Reed et al., 2022). As transformers are increasingly being leveraged for multi-task and multi-modal learning in domains with richer task structures, it is possible that these models may implicitly learn to decompose complex decisions into reusable, multi-level policies. In future work, we hope to explore these learning and generalization dynamics in transformer-based systems to understand the acquisition of task-conditioned, multi-level behavioral policies in structured environments.

### Acknowledgments

We would like to thank Andrew Nam, Satchel Grant, and members of the Stanford PDP lab for useful discussions. We also thank Andrew Lampinen and anonymous reviewers for helpful feedback on the manuscript.

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

# A  Additional Results

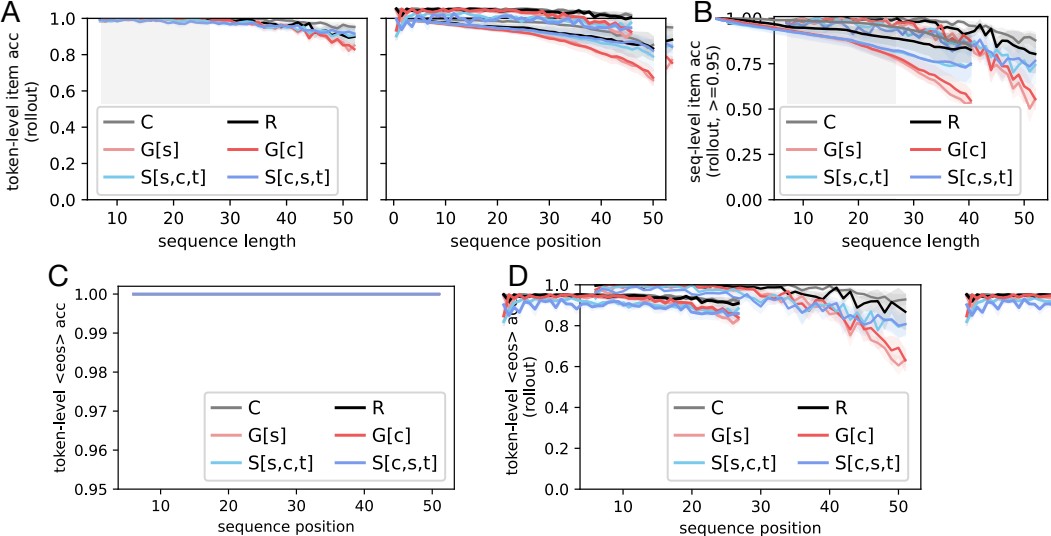

Figure S1: Multi-task models' accuracy under top1 rollout. **A**. Token-level rollout accuracy over sequence length and sequence positions. **B**. Sequence-level rollout accuracy where the models predict more than 95% tokens correct within sequence. In A and B, visualization as in Figure 3C. **C**. Six-task models predict the EOS token perfectly under teacher forcing. **D**. Prediction of the EOS token deteriorates under top1 rollout in six-task models. Results were aggregated across the five top-performing runs and taken from novel sequences within the training range.

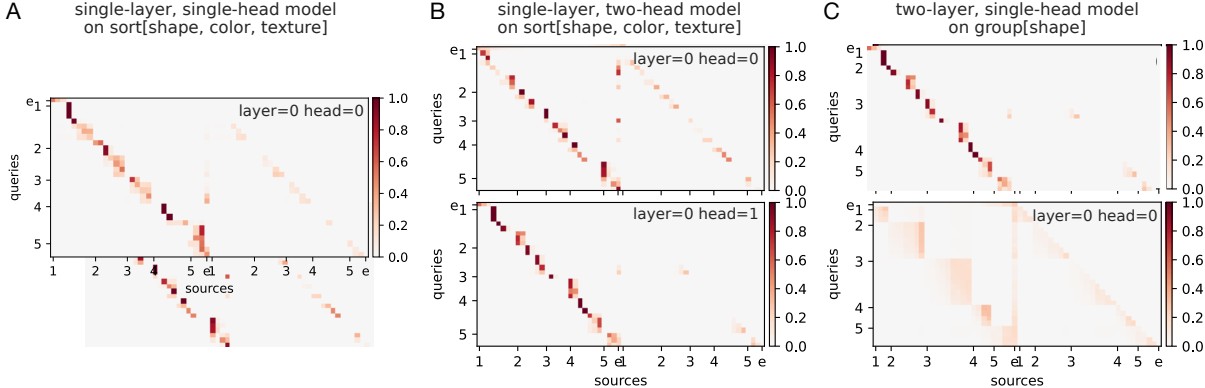

Figure S2: Attention maps from other single-task models. Visualization as in Figure 4B. Number labels indicate the beginning of each shape group. Label e indicates the EOS token.

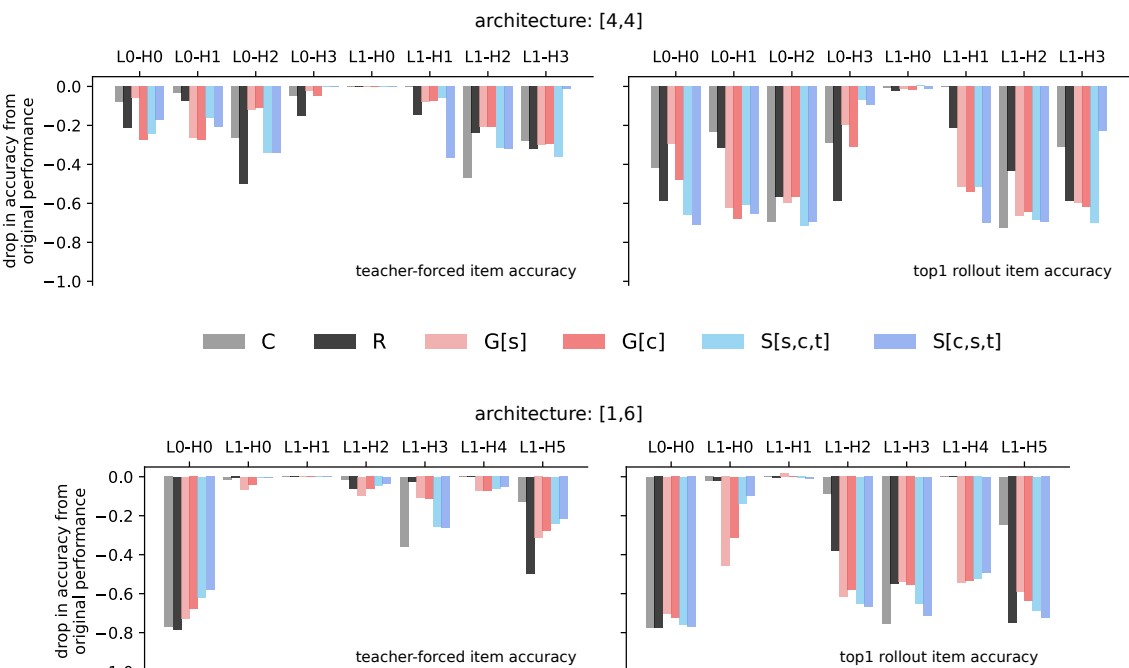

Figure S3: Attention ablation for two other top-performing models.

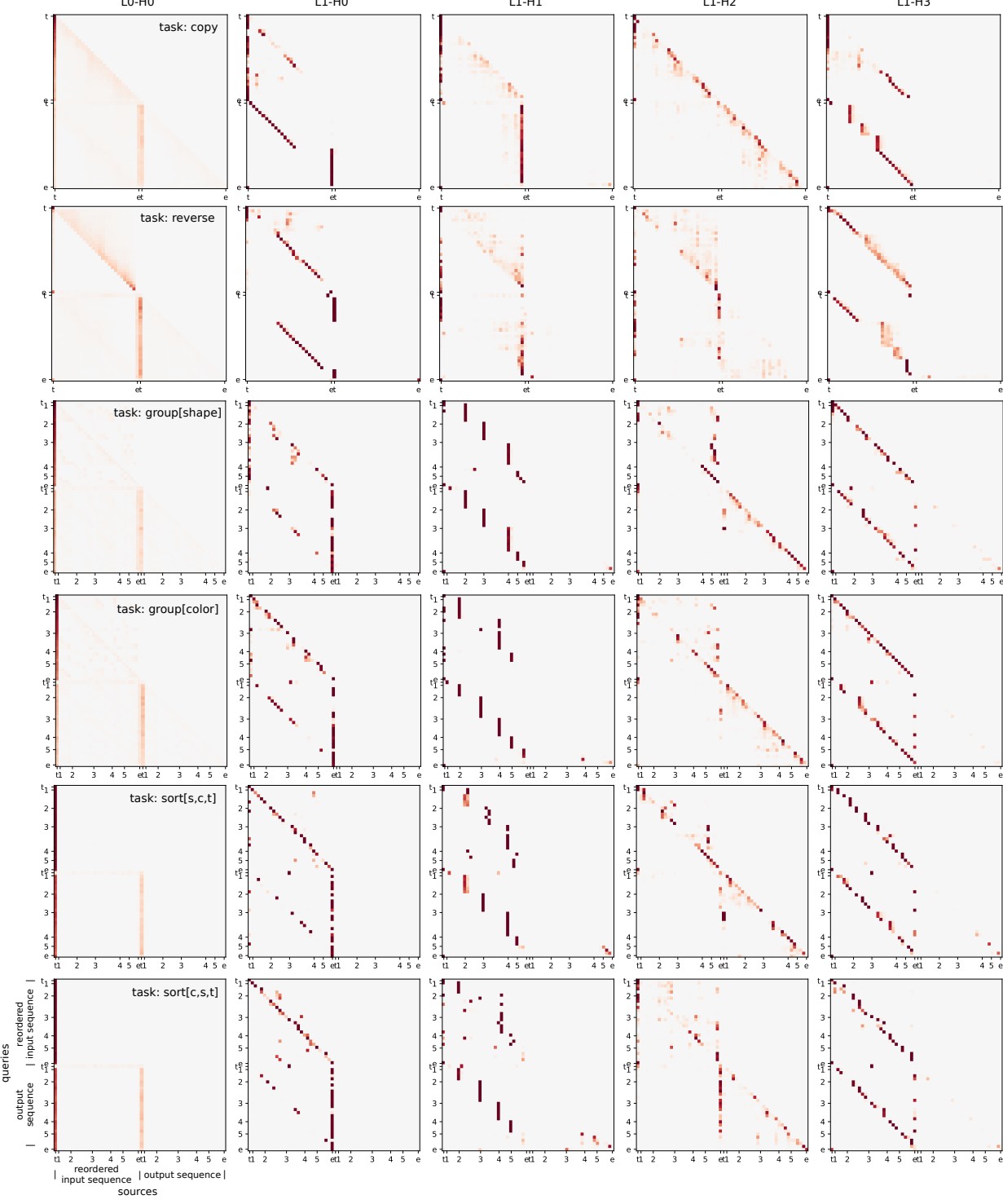

Figure S4: Full attention maps for an example generalization sequence (for the top-performing multi-task model with [1,4] architecture). Each row shows the attention maps for one task, and each column corresponds to a single attention head denoted by the layer and head index. Tokens corresponding to the input sequence were reorderd to match their order in the output sequence for visualization purposes. Label t indicates the task token. Label e indicates the EOS token. In the bottom four rows, number labels ranging from 1-5 mark the beginning of the items in each first-level feature group (shape or color).

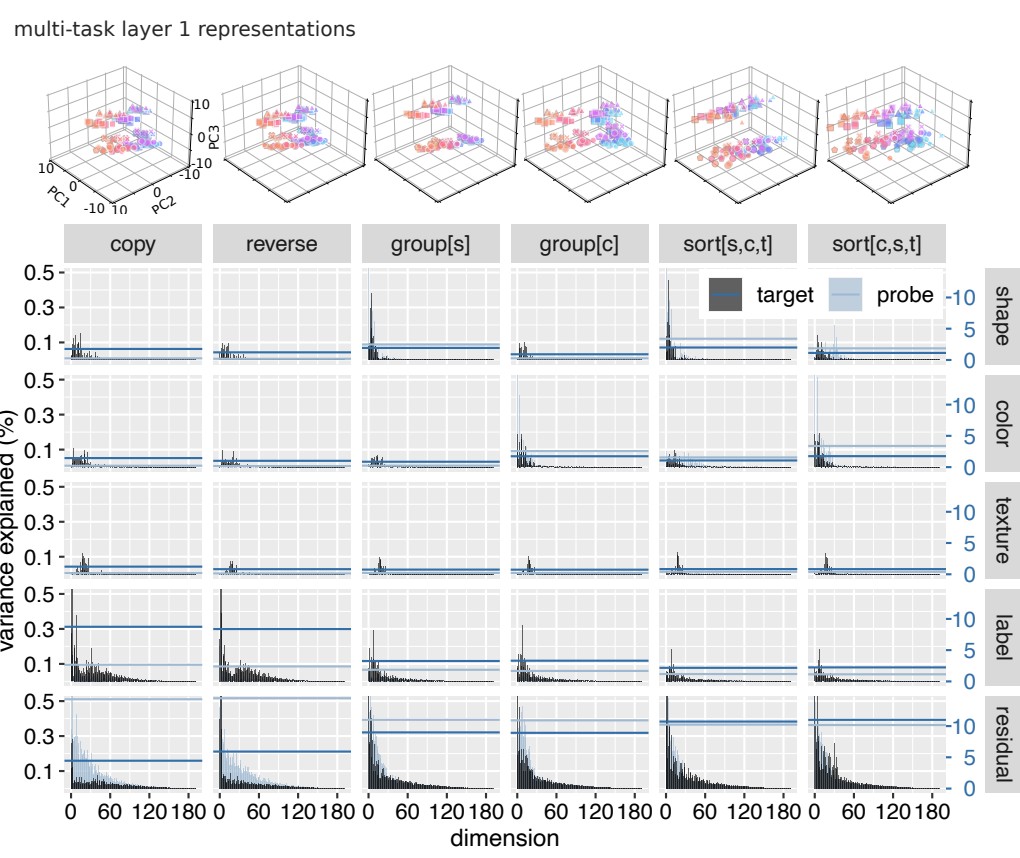

Figure S5: Multi-task model's layer 1 representations. Top, the lower-dimensional projection when representations are viewed as target items. Bottom, the variance-explained profile when representations are viewed as probe or target items. Blue lines indicate the total variance explained in a category across dimensions.

## B  Hyperparameters

Table S1: Model and experiment hyperparameters.

| Hyperparameter | Single-task learning | Six-task learning |
|---|---|---|
| Number of layers | 1 or 2 | 2 |
| Number of attention heads | (see paper) | |
| Embedding dimension | 128 (two-layer) or 184 (single-layer) | 192 |
| MLP hidden layer size | 64 | |
| Activation function | ReLU | |
| Batch size | 128 | |
| Training teacher forcing rate | 1.0 | |
| Optimizer | Adam | |
| Learning rate | $10^{-4}$ | $5 \cdot 10^{-4}$ |

## C  Quantitative Performance

Table S2: Token-level item and label accuracy across 54k generalization sequences in single-task models. Mean and standard deviation across four random seeds are shown for each architecture. Model architectures are denoted by the number of attention heads in each layer.

| task | architecture | position encoding | item prediction | label prediction |
|---|---|---|---|---|
| C | [1] | label | 99.03±0.41 | 100.00±0.00 |
| R | [1] | label | 99.75±0.21 | 100.00±0.00 |
| S[s,c,t] | [1] | label | 76.49±2.99 | 51.75±5.35 |
| S[s,c,t] | [2] | label | 91.01±2.76 | 78.02±5.89 |
| S[s,c,t] | [1,1] | label | 99.60±0.07 | 98.41±0.23 |
| G[s] | [1,1] | label | 98.41±0.32 | 99.31±0.16 |
| S[s,c,t] | [1,1] | sinusoidal | 57.04±10.75 | 16.50±6.52 |
| S[s,c,t] | [1,1] | learnable | 73.15±11.65 | 40.51±14.28 |

Table S3: Token-level prediction accuracy across 75k generalization sequences in six-task models (including 500 unique sequences for each length evaluated within each task). Mean and standard deviation across four random seeds are shown for each architecture. Model architectures are denoted by [# heads in the first layer, # heads in the second layer].

(a) Equal number of attention heads in each layer.

| Tasks | | [1,1] | [2,2] | [3,3] | [4,4] | [6,6] |
|---|---|---|---|---|---|---|
| all | item | 83.38±13.05 | 97.04±0.88 | 98.55±0.46 | 98.79±0.94 | **99.11±0.32** |
| | label | 87.21±7.86 | 96.46±1.24 | 97.98±0.42 | 98.68±0.84 | **99.17±0.32** |
| C | item | 89.28±14.44 | 97.57±1.19 | 98.81±0.77 | 98.98±0.95 | **99.13±0.97** |
| | label | 97.39±0.84 | 98.59±0.99 | 99.46±0.39 | 99.64±0.34 | **99.79±0.17** |
| R | item | 64.30±24.46 | 96.95±1.21 | 98.95±0.45 | 98.69±0.94 | **99.20±0.60** |
| | label | 94.56±8.26 | 98.53±0.80 | 99.64±0.16 | 99.39±0.29 | **99.73±0.19** |
| G[s] | item | 86.47±13.51 | 96.51±0.93 | 98.42±0.66 | 98.52±1.15 | **98.54±0.47** |
| | label | 91.54±8.30 | 97.83±1.07 | 99.07±0.33 | **99.23±0.52** | 99.05±0.30 |
| G[C] | item | 86.30±13.39 | 96.72±0.80 | 98.28±0.65 | 98.45±1.13 | **98.62±0.47** |
| | label | 91.69±7.79 | 97.93±1.06 | 98.98±0.19 | 99.21±0.46 | **99.22±0.24** |
| S[s,c,t] | item | 86.97±8.80 | 97.28±1.00 | 98.48±0.29 | 99.01±0.82 | **99.59±0.25** |
| | label | 74.75±11.47 | 93.21±2.23 | 95.53±1.01 | 97.09±1.99 | **98.62±0.82** |
| S[c,s,t] | item | 86.77±8.63 | 97.20±0.95 | 98.37±0.33 | 99.08±0.70 | **99.62±0.13** |
| | label | 73.57±10.59 | 92.73±2.05 | 95.23±0.90 | 97.53±1.71 | **98.65±0.54** |

(b) Multi-head attention in the first layer.

| Tasks | | [2,1] | [3,1] | [4,1] | [6,1] |
|---|---|---|---|---|---|
| all | item | 69.45±17.65 | 68.05±19.27 | 77.86±22.15 | **97.76±0.16** |
| | label | 79.95±8.53 | 78.35±11.60 | 86.53±11.67 | **97.52±0.31** |
| C | item | 57.93±27.09 | 54.07±30.36 | 72.01±30.48 | **98.50±0.81** |
| | label | 97.66±1.64 | 95.18±3.28 | 98.48±1.10 | **99.61±0.22** |
| R | item | 57.97±26.94 | 55.33±29.52 | 70.49±32.68 | **98.78±0.46** |
| | label | 95.74±3.34 | 94.90±3.50 | 97.97±2.17 | **99.56±0.23** |
| G[s] | item | 70.16±16.79 | 71.29±16.08 | 79.20±19.34 | **96.85±0.23** |
| | label | 80.56±11.79 | 79.37±12.51 | 87.82±12.52 | **98.96±0.26** |
| G[C] | item | 70.33±16.54 | 70.99±16.21 | 78.86±19.18 | **96.50±0.47** |
| | label | 79.18±12.71 | 78.71±12.82 | 87.28±12.69 | **98.82±0.32** |
| S[s,c,t] | item | 79.94±11.69 | 78.00±12.49 | 83.99±15.22 | **98.03±0.23** |
| | label | 62.76±13.85 | 61.17±19.50 | 76.69±18.13 | **94.17±0.74** |
| S[c,s,t] | item | 80.03±11.15 | 78.22±12.21 | 82.46±16.50 | **97.95±0.29** |
| | label | 64.28±12.87 | 61.27±18.82 | 71.27±24.19 | **94.08±0.82** |

(c) Multi-head attention in the second layer.

| Tasks | | [1,2] | [1,3] | [1,4] | [1,6] |
|---|---|---|---|---|---|
| all | item | 96.06±0.24 | 98.65±0.80 | **99.40±0.29** | 99.29±0.17 |
| | label | 95.48±0.30 | 97.37±0.89 | **99.21±0.43** | 99.12±0.36 |
| C | item | 96.86±0.57 | 99.10±0.74 | **99.46±0.34** | 99.22±0.52 |
| | label | 97.78±0.71 | 99.62±0.24 | **99.85±0.08** | 99.59±0.55 |
| R | item | 95.99±0.59 | 98.82±1.08 | 99.47±0.37 | **99.56±0.14** |
| | label | 98.09±0.52 | 99.55±0.39 | **99.83±0.04** | 99.82±0.09 |
| G[s] | item | 96.12±0.38 | 98.28±1.00 | **98.98±0.43** | 98.96±0.16 |
| | label | 97.70±0.42 | 98.77±0.66 | **99.29±0.27** | 99.25±0.07 |
| G[C] | item | 95.82±0.72 | 98.53±0.76 | **99.17±0.34** | 98.78±0.40 |
| | label | 97.51±0.51 | 99.06±0.42 | **99.51±0.18** | 99.13±0.32 |
| S[s,c,t] | item | 95.86±0.77 | 98.51±0.72 | **99.68±0.26** | 99.59±0.13 |
| | label | 91.12±0.83 | 93.54±2.44 | 98.40±1.16 | **98.41±0.89** |
| S[c,s,t] | item | 95.72±0.54 | 98.70±0.71 | **99.64±0.28** | 99.63±0.29 |
| | label | 90.74±0.66 | 93.76±2.15 | 98.36±1.05 | **98.54±1.10** |

