# OpenReview forum: "Representations and Computations in Transformers that Support Generalization on Structured Tasks"
_TMLR — Accepted by TMLR_

### Review · Reviewer_kxxZ · 2023-07-03

**Summary Of Contributions:**

This paper explores how a generative Transformer model discovers solutions as it learns to solve algorithmic tasks. Their results provide insights into how attention layers support structured computation within and across tasks, including how they learn generalizable layer-wise features.

**Audience:**

Yes

**Broader Impact Concerns:**

No ethical concerns.

**Claims And Evidence:**

Yes

**Requested Changes:**

I would like a better justification on why the proposed random label-based encoding can achieve the claimed benefit of "reliably communicate relative position information if ... in ascending order". The experiment results are positive, but I find it hard to be convincing unless the authors can show some good thought experiments. This may be a presentation issue: e.g., I am not sure if the sampled labels are always in ascending order. (If yes, it seems more reasonable to me since their embeddings may learn the notion like "2 is before 5 and 9 whatever" but it needs a good clarification from the authors.)

Can you explain how the random label technique is connected to the other parts of this paper?

**Strengths And Weaknesses:**

Strengths

This paper addresses a very interesting and potentially impactful question.

The authors carried out extensive experiments. Most claims are supported. Some results (e.g., multi-task model) are very interesting.

The paper is overall well-structured and some visualizations (e.g., Fig-1) are good.


Weakness

The main issue of this paper is that its conclusions seems anecdotal since the data and tasks are synthetic. It is not clear why the conclusions may generalize to more broad settings like text.

The paper is overall clear but vague/unclear at some key parts.

E.g., the reason why random-label should work is not sufficiently justified, see Requested Changes.

E.g., in 3.2, it is not clear why certain conclusions are drawn such as "multiple layers strongly benefit..."  The legends of mentioned figures are not defined (please correct me if I missed them) such as [1,1] and [2] in Fig-3.

It is not clear to me why the random label technique is connected to the other parts of the paper.

---

> ### Author Response · Authors · 2023-07-26
> **Response to Reviewer kxxZ**
>
> Thank you for your helpful feedback and we appreciate that you find the work interesting and the paper well-structured. Please see below for our response to the noted weakness and requested changes:
>
> > The main issue of this paper is that its conclusions seems anecdotal since the data and tasks are synthetic. It is not clear why the conclusions may generalize to more broad settings like text.
>
> We agree that further research is needed to confirm the relevance of our specific findings on larger models learning more complex tasks such as natural language tasks. Like Reviewer zzw5, we think studying controlled and synthetic tasks allowed us to dig deeper into the phenomena and more thoroughly understand the model, which provides a starting place for future work. Ultimately, like others, we are interested in understanding if similar conclusions can extend to more complex settings. We have extensively expanded the discussion on this in the Discussion section to highlight our thoughts on the broader implications of this work.
>
> > in 3.2, it is not clear why certain conclusions are drawn such as "multiple layers strongly benefit..." The legends of mentioned figures are not defined (please correct me if I missed them) such as [1,1] and [2] in Fig-3.
>
> Thank you for raising this potential confusion. We have highlighted and clarified the legend in the main text (page 5, section 3.2) and in Figure 3 in the revised manuscript.
>
> > the reason why random-label should work is not sufficiently justified/a better justification on why the proposed random label-based encoding can achieve the claimed benefit
>
> Yes, you’re correct that the sampled labels are always in ascending order in our experiments. When they appear in ascending order, they communicate sequence order because each successive token is paired with a larger integer label. There can be situations where these labels don’t need to be sorted in ascending order, e.g. when token order is irrelevant for a given task, as was the case in the sort tasks in our task suite (though in our experiments we used ascending labels for all tasks). We have modified the introduction of this encoding method on page 3 to clarify these points.
>
> > Can you explain how the random label technique is connected to the other parts of this paper?
>
> We see the random label technique as an integral part of the paper because how the model represents the sequence order is a fundamental part of applying transformers to any problem. In our case, different ways of encoding sequence order have a significant impact on the model’s ability to systematically generalize, and, importantly, our random label method can help in-distribution learning in transformers when the length of sequences in the training set varies. Thus, using the label encoding method was key to encouraging the model to develop a more structured solution to these tasks, and supports all the subsequent analyses on model computation and representation.

---

### Review · Reviewer_rtG2 · 2023-07-06

**Summary Of Contributions:**

This work presents an in-depth analysis of transformer models trained to perform algorithmic tasks, incorporating a novel PCA-based analysis method to better understand how representations are transformed to support single- and multi-task learning. A novel positional encoding scheme is also introduced that supports better systematic generalization to sequences longer than those observed during training.

**Audience:**

Yes

**Broader Impact Concerns:**

There are no discernible ethical issues related to this work.

**Claims And Evidence:**

Yes

**Requested Changes:**

I have a few comments and suggestions listed above that may improve the paper further, but I think the paper is already in good shape.

**Strengths And Weaknesses:**

Strengths:

- The novel positional encoding scheme is an interesting proposal, and displays a promising ability to support systematic generalization to longer sequences.
- Task space is simple yet sufficiently diverse to explore how models might learn to compose operations in support of structured multi-task reasoning.
- Extremely thorough analyses yielding many insights concerning how transformers learn to perform algorithmic tasks. The role of inductive biases (especially attention and multi-layer composition of operations) is intuitively and effectively conveyed by the analyses and text.
- An interesting analysis method is presented using PCA to show how representations are transformed based on task demands. This approach seems likely to be useful in many other contexts.

Comments:

- Are there any drawbacks to the proposed positional encoding scheme? In particular, I'm wondering if this approach makes it harder to make precise positional judgments. For instance, how would this approach fare on a task such as 'given a sequence of objects, how far apart are objects A and B?'
- As an alternative encoding scheme, have the authors considered simply randomly scaling the positional indices? This seems like it would convey the same benefit of generalization to longer sequences, but would also preserve the precise metric relations between positions.
- The proposed encoding scheme is clearly beneficial in terms of generalization to longer sequences, but I was somewhat surprised that both learned and sinusoidal positional codes also underperformed on the training set. Do the authors have some sense for why these methods also struggled in an iid setting?

---

> ### Author Response · Authors · 2023-07-26
> **Response to Reviewer rtG2**
>
> Thank you for your positive comments and helpful suggestions. We're happy to hear that you find the paper interesting and clear. We respond to your comments below:
>
> > Are there any drawbacks to the proposed positional encoding scheme? In particular, I'm wondering if this approach makes it harder to make precise positional judgments. For instance, how would this approach fare on a task such as 'given a sequence of objects, how far apart are objects A and B?'
>
> We definitely acknowledge that this method does not preserve absolute distance information between items, and we have clarified this in our revision (page 11). It may be possible to combine the label encoding scheme and other distance-preserving position encoding schemes (e.g., learnable position encoding) to allow the model to both achieve length generalization and remain sensitive to true token distances.
>
> > As an alternative encoding scheme, have the authors considered simply randomly scaling the positional indices? This seems like it would convey the same benefit of generalization to longer sequences, but would also preserve the precise metric relations between positions.
>
> Thank you for this interesting proposal. It certainly seems that randomly scaling positional indices can achieve some of the same benefits of our random label encoding method. We are not certain we fully understand your exact proposal, and hope you will clarify. If we understood it correctly, it seems that it would require sampling different scaling factors for different sequence instances to fully support length generalization, but random scaling factors would similarly distort the absolute position distance of the items. In future work, it would be interesting to compare whether scaling positional indices (so that tokens are equidistant within sequences) achieve the same level of learning and generalization benefits as our random label encoding method.
>
> > The proposed encoding scheme is clearly beneficial in terms of generalization to longer sequences, but I was somewhat surprised that both learned and sinusoidal positional codes also underperformed on the training set. Do the authors have some sense for why these methods also struggled in an iid setting?
>
> We suspect that this relates to some positions being over-represented across sequences of varying lengths. During training, position encodings 1-5 always appear for each sequence but position encodings 20-25 appear much less commonly. This can hinder the model’s ability to develop solutions that handle position encodings for infrequently encountered positions as well as they do for frequent ones. We note that the learning curves for both learned and sinusoidal positional encodings may continue to increase and eventually learn to sort the training sequences, but label encoding evens out the probability of each label appearing during training, which helps the model learn a structured solution much faster.

---

> > ### Comment · Reviewer_rtG2 · 2023-08-03
> > **Reply**
> >
> > Thanks to the authors for this response. Just to clarify my proposal about random scaling: the idea is that this would at least preserve the relative distances between tokens, even if it doesn't preserve the absolute distances. But I agree that it is not necessary to explore this idea in the present work. All of my concerns have been addressed.

---

### Review · Reviewer_zzw5 · 2023-07-15

**Summary Of Contributions:**

The authors train Transformers to perform a range of synthetic sequence-manipulation tasks (e.g., reversing the sequence, or sorting a sequence based on features of the iterms). They then perform extensive and fine-grained analyses of the representations and processing of the trained models to understand how they have learned to perform these tasks. In more detail, the contributions include:
- The set of tasks that models are trained on, as a useful setting for performing model analysis
- A new positional encoding scheme that improves length generalization on these tasks. (As the authors note, another paper has concurrently introduced the same scheme, but the authors describe how their analyses of the scheme are complementary.)
- A determination of the minimal number of heads and layers that are needed for Transformers to perform different individual tasks, or combinations of tasks in a multitask setting
- Attention-based analyses showing how models use attention for different targeted purposes in different layers to compute the information needed to solve the task in a structured, multi-step way.
- Analysis of attention head sharing, task embedding similarity, and cross-task decodability transfer of information supporting the view that multitask models have learned to partially but not completely share information across tasks
- PCA analyses showing how representations induced for different tasks reflect the different types of information that are useful for the task in question (e.g., color is prioritzed in a color-based sorting task, while shape is prioritized in a shape-based sorting task)


**Audience:**

Yes

**Claims And Evidence:**

Yes

**Requested Changes:**

None of these are necessary for securing my recommendation; they are just suggestions that could clarify some aspects of the paper.
- RC1: I found the term “label encoding” to be unintuitive. I typically think of “label” as meaning “something that captures some information about an item” (e.g., a part-of-speech tag), whereas these labels are random and arbitrary. Perhaps another name would be clearer, such as “random ascending positions”?
- RC2: The main contributions (1, 2, and 3 at the end of section 1) did not make much sense to me until I had read the rest of the paper. Given that this paper is dense, I expect that readers would really benefit from having a clearer summary at the start so that they can get the main points more clearly without diving into the details. Therefore, I would recommend expanding on these points a bit. Specifically:
    - a. Under point 1, I recommend briefly summarizing the structure of the label based encoding
    - b. Under point 2, I would recommend adding a sentence that says “For example, …” and then gives 1 or 2 examples of specific results, in order to give readers an intuition about what sorts of “systematic decomposition” and “exploitation of shared structures across tasks” are being discussed. E.g., you could say “For example, we find that 2-layer transformers implement a 2-step algorithm with the attention patterns in each layer performing one step of the algorithm; and we find that, in multitask models, each attention head is used roughly equally by all tasks.”
    - c. Under point 3, similar to point 2, I would recommend adding a “For example…” to give readers intuition about what it means to “re-weight task-relevant information”
- RC3: Figure 3B: In the caption, maybe mention that models are arranged in ascending order of performance? I assumed at first that it was in ascending order of size, so I missed that 1,6 is to the left of 1,4 in the bottom row.
- RC4: Figure 4B, x-axis labels and y-axis albels: the “e” labels are hard to see because it touches the circle - could they be separated a bit? The same comment also applies to “t” labels in Figure 5C (for x-axes and y-axes)
In the middle of page 6, it says “reflects an advantage of the two-layer models in providing an inductive bias favoring a structured, multi-stage solution.” I slightly disagree with this characterization: It seems more like this task requires a multi-stage solution, such that any model with the capacity to perform a multi-stage solution would learn to implement one. Thus, it may not be that 2 layers gives an inductive bias for multi-stage solutions, but rather that it gives the capacity necessary to develop a multi-stage solution. In order to say that it has a bias, I think the proper type of evidence would be to show that it learns a multi-stage solution even when the task does not demand it; when the task does demand it (as it seems to do here), then the multi-stage nature seems likely to be driven by the task rather than by the model’s inductive biases.
- RC5: When I first read the first paragraph in Section 3.4, I was confused and didn’t understand it. It may be worthwhile to say somewhere in that paragraph “As we will discuss in more detail in the next few paragraphs…” to signal to readers that this paragraph is just a summary. (My confusion came from the fact that I thought the paragraph was standalone, so I was confused about why it didn’t give numbers or explanations to support the statements about what the results reveal).
- RC6: When it says “The results show that such a classifier was above chance“ (on page 9), it might be worth it to remind readers of what chance was
- RC7: A related paper about RNNs that you might enjoy is: Processing of nested and cross-serial dependencies: An automaton perspective on SRN behaviour. Kirov and Frank 2012. https://www.tandfonline.com/doi/pdf/10.1080/09540091.2011.641939.


**Strengths And Weaknesses:**

Overall, I found this to be a very strong paper - I would like to thank the authors for creating such interesting and illuminating work!

Strengths:
- S1: The experimental setup is careful and well-motivated. The authors have chosen tasks that are simple enough to be interpreted yet complex enough to reveal interesting insights about how Transformers operate. This setting may be useful for future authors wishing to conduct mechanistic interpretability work.
- S2: The results make important progress toward the goal of understanding “black-box” neural networks that are notoriously hard to understand - a goal that the community is increasingly recognizing as an important one.
- S3: The paper uses an impressive range of analysis techniques (e.g., attention analyses, attention ablations, probing classifiers, and more). There are 2 reasons why this is a major positive: First, it provides converging evidence for the authors’ conclusions. Second, these analysis techniques (like the tasks/datasets) may be useful for future work.
- S4: The paper is clear and well-written. Although there is a lot in it, the authors have explained everything very concisely, so I don’t think it could be condensed more without weakening the paper. There were also several occasions where I had a question that was immediately answered in the next paragraph - a sign that the authors have carefully thought through their paper to anticipate any confusion!
- S5: I really enjoyed the figures - they have lots of content arranged in a careful way, so it was fun to dive into them.
- S6: The paper seems like a great fit for TMLR: It has much more content than a typical conference paper, so the journal format is perfect for being able to include the extensive, detailed analysis.

Weaknesses: There were no major weaknesses that I identified. The one partial weakness is that there is so much in the paper that it is a bit dense, and it requires some focus to process everything. But I am fine with that, and I think that the authors did a good job of balancing conciseness with clarity (see S4). Some readers may also view it as a weakness that the paper only uses synthetic tasks, but I do not think this is a weakness - in fact, I think it is a strength because it makes it possible to understand the phenomena thoroughly, paving the way for analysis of more complicated phenomena (see S1).

---

> ### Author Response · Authors · 2023-07-26
> **Response to Reviewer zzw5**
>
> Thank you for your in-depth review of our paper and for your detailed suggestions. We are very glad to hear that you enjoyed diving into the details of this work and found the results interesting and illuminating. We appreciate all of the positive feedback even in statements of a partial weakness. Please see our response to the noted weakness/requested changes below:
>
> > The one partial weakness is that there is so much in the paper that it is a bit dense, and it requires some focus to process everything.
>
> We aspire to ensure that the paper is accessible, so we are taking the opportunity to revise as an opportunity to ensure everything is fully clear.  We have added additional signposting in the presentation, following your suggestions made in requested changes.
>
> > RC1: I found the term “label encoding” to be unintuitive. I typically think of “label” as meaning “something that captures some information about an item” (e.g., a part-of-speech tag), whereas these labels are random and arbitrary. Perhaps another name would be clearer, such as “random ascending positions”?
>
> Thank you for sharing this. In our revised paper, we have clarified that the labels are randomly sampled arbitrary labels, and have clarified how labels can be completely decoupled from positions when used in random order (see page 3). The concurrent related work (Ruoss et al., 2023) named the related method randomized positional encodings, and we intend to stick to “label encoding” to reflect our slightly different interpretation of the method in that it can be used more flexibly than in ascending order.
>
> > RC2 ~ RC6
>
> Thank you for flagging these points of confusion and for providing these detailed suggestions. We went ahead and updated all of the corresponding sections and figures to provide these clarifications.
>
> > In the middle of page 6, it says “reflects an advantage of the two-layer models in providing an inductive bias favoring a structured, multi-stage solution.” I slightly disagree with this characterization
>
> Thank you for pointing this out. We agree that two-layer models providing the capacity to discover a multi-stage solution is a more accurate characterization and have updated the manuscript accordingly at the bottom of page 6.
>
> > RC7: A related paper about RNNs that you might enjoy
>
> Thank you for the pointer to this paper. We will read it with interest. The types of dependencies described in the title are unlike those we have considered here, and could be an interesting target for a future extension of our work.

---

### Author Response · Authors · 2023-07-26
**General Response**

We thank all reviewers for taking the time to review our submission and for their positive and constructive feedback. We have revised the paper to enhance/clarify the points raised in the reviews, and we provide more specific responses to each reviewer below.

Two common themes from the reviews concern clarification around the label-based encoding method and the generalization of our findings on synthetic tasks to more complex settings. We’re glad to see that the reviewers see the value in the label-based encoding method we propose, and we have clarified the method in both the revised manuscript and in our response below. We have also expanded our discussion on the broader implications of our work, including the implications the variance-based representation analysis may have on mechanistic understanding of more complex models/naturalistic tasks in light of related work. Like Reviewer zzw5, we think that using the synthetic tasks to push the boundary of mechanistic understanding with small models can yield insights into the kinds of computations and representations we may expect to be at play in complex models. In addition, these settings provide a test ground for devising useful analysis tools. We hope that the revised manuscript and our response below address the concerns raised and we thank all reviewers again for their enthusiasm about our work.

---

### Decision · Action_Editors · 2023-08-22

**Recommendation:** Accept as is

**Comment:**

All reviewers recommend acceptance of this paper and the AE agrees. Modifications made during the reviewing process improved the clarity of the proposed methodology.

**Audience:**

All reviewers agree there is likely an interested audience at TMLR for this work and the AE agrees. Building intuitions and techniques for mechanistic understanding in synthetic tasks can provide useful insights and lay the foundations for more real-world applications.

**Claims And Evidence:**

All reviewers agree the claims and evidence were sufficient and the AE agrees.

**Resubmission Of Major Revision:**

The authors may consider submitting a major revision at a later time.